# AdaGAN: Adaptive GAN for Many-to-Many Non-Parallel Voice Conversion

## Abstract

Voice Conversion (VC) is a task of converting perceived speaker identity from a source speaker to a particular target speaker. The earlier approaches in the literature primarily find a mapping between the given source-target speaker-pairs. Developing mapping techniques for many-to-many VC using non-parallel data, including zero-shot learning, remains less explored areas in VC. Most of the many-to-many VC architectures require training data from all the target speakers for whom we want to convert the voices. In this paper, we propose a novel style transfer architecture, which can also be extended to generate voices even for target speakers whose data were not used in training (i.e., case of zero-shot learning). In particular, we propose Adaptive Generative Adversarial Network (AdaGAN), new architectural training procedure that helps in learning normalized speaker-independent latent representation, which will be used to generate speech with different speaking styles in the context of VC. We compare our results with the state-of-the-art StarGAN-VC architecture. In particular, the AdaGAN achieves 31.73%, and 10.37% relative improvement compared to the StarGAN in MOS tests for speech quality and speaker similarity, respectively. The key strength of the proposed architectures is that it yields these results with less computational complexity. AdaGAN is 88.6% less complex than StarGAN-VC in terms of FLoating Operation Per Second (FLOPS), and 85.46% less complex in terms of trainable parameters.

## 1 Introduction

Language is the core of civilization, and speech is the most powerful and natural form of communication. Human voice mimicry has always been considered as one of the most difficult tasks since it involves understanding of the sophisticated human speech production mechanism (Eriksson & Wretling (1997)) and challenging concepts of prosodic transfer (Gomathi et al. (2012)). In the literature, this is achieved using Voice Conversion (VC) technique (Stylianou (2009)). Recently, VC has gained more attention due to its fascinating real-world applications in privacy and identity protection, military operations, generating new voices for animated and fictional movies, voice repair in medical-domain, voice assistants, etc. Voice Conversion (VC) technique converts source speaker's voice in such a way as if it were spoken by the target speaker. This is primarily achieved by modifying *spectral* and *prosodic* features while retaining the linguistic information in the given speech signal (Stylianou et al. (1998)). In addition, Voice cloning is one of the closely related task to VC (Arik et al. (2018)). However, in this research work we only focus to advance the Voice Conversion.

With the emergence of deep learning techniques, VC has become more efficient. Deep learning-based techniques have made remarkable progress in parallel VC. However, it is difficult to get parallel data, and such data needs alignment (which is a arduous process) to get better results. Building a VC system from non-parallel data is highly challenging, at the same time valuable for practical application scenarios. Recently, many deep learning-based style transfer algorithms have been applied for non-parallel VC task. Hence, this problem can be formulated as a style transfer problem, where one speaker's style is converted into another while preserving the linguistic content as it is. In particular, Conditional Variational AutoEncoders (CVAEs), Generative Adversarial Networks (GANs) (proposed by Goodfellow et al. (2014)), and its variants have gained significant attention in non-parallel VC. However, it is known that the training task for GAN is hard, and the convergence property of GAN is fragile (Salimans et al. (2016)). There is no substantial evidence that the gen-

erated speech is perceptually good. Moreover, CVAEs alone do not guarantee distribution matching and suffers from the issue of over smoothing of the converted features.

Although, there are few GAN-based systems that produced state-of-the-art results for non-parallel VC. Among these algorithms, even fewer can be applied for many-to-many VC tasks. At last, there is the only system available for zero-shot VC proposed by Qian et al. (2019). Zero-shot conversion is a technique to convert source speaker's voice into an unseen target speaker's speaker via looking at a few utterances of that speaker. As known, solutions to a challenging problem comes with trade-offs. Despite the results, architectures have become more complex, which is not desirable in real-world scenarios because the quality of algorithms or architectures is also measured by the training time and computational complexity of learning trainable parameters (Goodfellow et al. (2016)).

Motivated by this, we propose computationally less expensive Adaptive GAN (AdaGAN), a new style transfer framework, and a new architectural training procedure that we apply to the GAN-based framework. In AdaGAN, the generator encapsulates Adaptive Instance Normalization (AdaIN) for style transfer, and the discriminator is responsible for adversarial training. Recently, StarGAN-VC (proposed by Kameoka et al. (2018)) is a state-of-the-art method among all the GAN-based frameworks for non-parallel many-to-many VC. AdaGAN is also GAN-based framework. Therefore, we compare AdaGAN with StarGAN-VC for non-parallel many-to-many VC in terms of naturalness, speaker similarity, and computational complexity. We observe that AdaGAN yields state-of-the-art results for this with almost 88.6% less computational complexity. Recently proposed AutoVC (by Qian et al. (2019)) is the only framework for zero-shot VC. Inspired by this, we propose AdaGAN for zero-shot VC as an independent study, which is the first GAN-based framework to perform zero-shot VC. We reported initial results for zero-shot VC using AdaGAN.The main contributions of this work are as follows:

- We introduce the concept of latent representation based many-to-many VC using GAN for the first time in literature.
- We show that in the latent space content of the speech can be represented as the distribution and the properties of this distribution will represent the speaking style of the speaker.
- Although AdaGAN has much lesser computation complexity, AdaGAN shows much better results in terms of naturalness and speaker similarity compared to the baseline.

## 2 RELATED WORK

Developing a non-parallel VC framework is challenging task because of the problems associated with the training conditions using non-parallel data in deep learning architectures. However, attempts have been made to develop many non-parallel VC frameworks in the past decade. For example, Maximum Likelihood (ML)-based approach proposed by Ye & Young (2006), speaker adaptation technique by Mouchtaris et al. (2006), GMM-based VC method using Maximum a posteriori (MAP) adaptation technique by Lee & Wu (2006), iterative alignment method by Erro et al. (2010), Automatic Speech Recognition (ASR)-based method by Xie et al. (2016), speaker verification-based method using i-vectors by Kinnunen et al. (2017), and many other frameworks (Chen et al. (2014); Nakashika et al. (2014); Blaauw & Bonada (2016); Hsu et al. (2016); Kaneko & Kameoka (2017); Saito et al. (2018a); Sun et al. (2015); Shah et al. (2018b;c); Shah & Patil (2018); Biadsy et al. (2019)). Recently, a method using Conditional Variational Autoencoders (CVAEs) (Kingma & Welling (2013)) was proposed for non-parallel VC by (Hsu et al. (2016); Saito et al. (2018a)). Recently, VAE based method for VC was proposed, which also uses AdaIN to transfer the speaking style (Chou et al. (2019)). One powerful framework that can potentially overcome the weakness of VAEs involves GANs. While GAN-based methods were originally applied for image translation problems, these methods have also been employed with noteworthy success for various speech technology-related applications, we can see via architectures proposed by (Michelsanti & Tan (2017); Saito et al. (2018b); Shah et al. (2018a)), and many others. In GANs-based methods, Cycle-consistent Adversarial Network (CycleGAN)-VC is one of the state-of-the-art methods in the non-parallel VC task proposed by (Kaneko & Kameoka (2017)).

Among these non-parallel algorithms, a few can produce good results for non-parallel many-to-many VC. Recently, StarGAN-VC (Kameoka et al. (2018)) is a state-of-the-art method for the non-parallel many-to-many VC among all the GAN-based frameworks. Past attempts have been made

to achieve conversion using style transfer algorithms (Atalla et al. (2018); Chou et al. (2018); Qian et al. (2019)). The most recent framework is the AutoVC (proposed by Qian et al. (2019)) using style transfer scheme, the first and the only framework in VC literature which achieved state-of-the-art results in zero-shot VC.

## 3   APPROACH

### 3.1   PROBLEM FORMULATION

The traditional VC problem is being reformulated as a style transfer problem. Here, we assume $Z$ is a set of $n$ speakers denoted by $Z = \{Z_1, Z_2, ..., Z_n\}$, where $Z_i$ is the $i^{th}$ speaker, and $U$ is the set of $m$ speech utterances denoted by $U = \{U_1, U_2, ..., U_m\}$, where $U_i$ is the $i^{th}$ speech utterance. Now, probability density function (*pdf*) is generated for given $Z_i$, and $U_i$ denoted by $p_X(.|Z_i, U_i)$ via the stochastic process of random sampling from the distributions $Z_i$ and $U_i$. Here, $X_i \sim p_X(.|Z_i, U_i)$ can be referred as features of given $U_i$ with speaking style of $Z_i$.

The key idea is to transfer the speaking style of one speaker into another in order to achieve VC. For this, let us consider a set of random variables $(Z_1, U_1)$ corresponding to a source speaker, and $(Z_2, U_2)$ corresponding to a target speaker. Here, $U_1$ and $U_2$ are spoken by $Z_1$ and $Z_2$, respectively. Our goal is to achieve $p_{\hat{X}}(.|Z_2, U_1)$. Now, we want to learn a mapping function to achieve our goal for VC. Our mapping function is able to generate the distribution denoted by $\hat{X}_{Z_1 \rightarrow Z_2}$ with speaking style of $Z_2$ while retaining the linguistic content of $U_1$. Formally, we want to generate the pdf (i.e., $p_{\hat{X}_{Z_1 \rightarrow Z_2}}(.|Z_1, U_1, Z_2, U_2)$) to be close or equal to the $p_{\hat{X}}(.|Z_2, U_1)$. Accurately, our mapping function will achieve this property, as shown in eq. 1.

$$p_{\hat{X}_{Z_1 \rightarrow Z_2}}(.|Z_1, U_1, Z_2, U_2) = p_{\hat{X}}(.|Z_2, U_1). \tag{1}$$

Intuitively, we want to transfer the speaking style of $Z_2$ to the $Z_1$ while preserving the linguistic content of $U_1$. Therefore, converted voice is perceptually sound as if utterance $U_1$ were spoken by $Z_2$. With this, AdaGAN is also designed to achieve zero-shot VC. During zero-shot conversion, $U_1$ and $U_2$ can be seen or unseen utterances, and $Z_1$ and $Z_2$ can be seen or unseen speakers.

### 3.2   ADAPTIVE INSTANCE NORMALIZATION ($AdaIN$)

Our key idea for style transfer in VC revolves around the $AdaIN$. First, AdaIN was introduced for arbitrary style transfer in image-to-image translation tasks by Huang & Belongie (2017). In this paper, $AdaIN$ helps us to capture the speaking style and linguistic content into a single feature representation. $AdaIN$ takes features of a source speaker's speech (i.e., $X$) and sample features of the target speaker's speech (i.e., $Y$). Here, $x$ is a feature from the set $X$ related to the linguistic content of source speech, and $Y$ is features related to the speaking style of the target speaker. $AdaIN$ will map the mean and standard deviation of $X$ (i.e., $\mu_X$ and $\sigma_x$) in such a way that it will match with mean, and standard deviation of $Y$ (i.e., $\mu_Y$ and $\sigma_Y$). Mathematical equation of $AdaIN$ is defined as (Huang & Belongie (2017)):

$$AdaIN(x, Y) = \sigma_Y \left( \frac{x - \mu_X}{\sigma_X} \right) + \mu_Y. \tag{2}$$

From eq. (2), we can infer that *AdaIN* first normalizes $x$, and scales back based on mean and standard deviations of $y$. Intuitively, let's assume that we have one latent space which represents the linguistic content in the distribution and also contains speaking style in terms of the mean and standard deviation of the same distribution. To transfer the speaking style, we have adopted the distribution properties (i.e., its mean and standard deviation) of the target speaker. As a result, the output produced by *AdaIN* has the high average activation for the features which are responsible for style (y) while preserving linguistic content. $AdaIN$ does not have any learning parameters. Hence, it will not affect the computational complexity of the framework.

## 4   PROPOSED ADAGAN FRAMEWORK

In this Section, we discuss our proposed AdaGAN architecture in detail. we show that AdaIN helps the generator make speaking style transfer easy and efficient, and can achieve zero-shot VC. We present an intuitive and theoretical analysis for the proposed framework.

The AdaGAN framework consists of an encoder $En(.)$, a decoder $De(.)$, and a discriminator $Dis(.)$. Here, $En(.)$ encodes the input features of speech to the latent space, $De(.)$ generates the features of speech from the given latent space, and $Dis(.)$ ensures adversarial training. The style transfer scheme and training procedure are shown in Fig. 1.

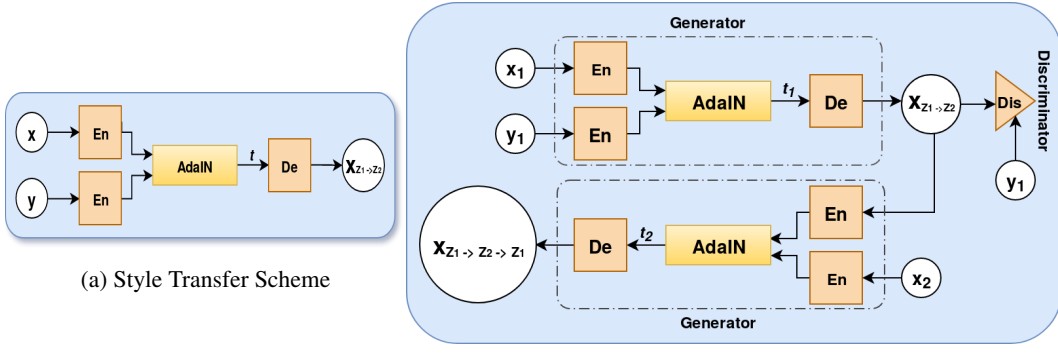

(a) Style Transfer Scheme

(b) Training Methodology

Figure 1: Schematic representation of proposed AdaGAN architecture.

### 4.1 PROPOSED STYLE TRANSFER SCHEME

Features of source speaker's speech (i.e., $x$), and any sample features of target speaker's speech (i.e., $y$), is taken as input to $En(.)$ to get the required latent space representations $S_x$ and $S_y$ as given in eq. 3. Now, $AdaIN$ is used to transfer distribution properties (i.e., its mean and standard deviation) of $S_y$ to $S_x$, and generate the single feature representation denoted by $t$ as per eq. 3. In the next step, we have used $De(.)$ to generate features of speech (i.e., $x_{Z_1 \to Z_2}$) from $t$. This entire process is illustrated via Fig. 1(a). This generated features $x_{Z_1 \to Z_2}$ contains the speaking style of target speaker via retaining the linguistic content of source speaker speech. We have encapsulated this style transfer algorithm into the generator of AdaGAN in order to improve the quality of $x_{Z_1 \to Z_2}$ via adversarial training.

$$S_x = En(x), \quad S_y = En(y), \quad t = AdaIN(S_x, S_y), \quad x_{Z_1 \to Z_2} = De(t). \tag{3}$$

### 4.2 TRAINING AND TESTING METHODOLOGY

We have applied a new training methodology in GAN-based framework. We have designed a training procedure based on non-parallel data in order to learn the mapping function for many-to-many as well as zero-shot VC. We know that the idea of transitivity as a way to regularize structured data has a long history. People have extended this concept into the training methodologies of deep learning architectures (Zhu et al. (2017); Kim et al. (2017)). In this paper, we have encapsulated the idea of transitivity via introducing the reconstruction loss along with adversarial training. The entire training procedure is illustrated in Fig. 1(b).

First, we randomly select the two speakers $Z_1$ and $Z_2$. Formally, we have two sets of random variables, $(Z_1, U_1, X)$ and $(Z_2, U_2, Y)$ corresponding to the source and target speaker, respectively. After this, we randomly select $x_1, x_2 \in p_X(.|Z_1, U_1)$, and $y_1, y_2 \in p_Y(.|Z_2, U_2)$.

During the training, VC is done from the source speaker ($Z_1$) to target speaker ($Z_2$) via style transfer scheme illustrated in Fig. 1(a). Using $x_1, y_1$, we transfer speaking style of speaker $Z_2$ to $Z_1$. From eq. (3), we can describe this procedure as shown in eq. (4).

$$S_{x_1} = En(x_1), \quad S_{y_1} = En(y_1), \quad t_1 = AdaIN(S_{x_1}, S_{y_1}), \quad x_{Z_1 \to Z_2} = De(t_1). \tag{4}$$

Now, using another sample of source speech (i.e., $x_2$), we have reconstructed the source speech features (i.e., $x_{Z_1 \to Z_2 \to Z_1}$) from the features of converted speech ($x_{Z_1 \to Z_2}$) in order to achieve better conversion efficiency. This procedure is described in eq. (5).

$$S_{x_2} = En(x_2), \quad S_{x_{Z_1 \to Z_2}} = En(x_{Z_1 \to Z_2}), \quad t_2 = AdaIN(S_{x_{Z_1 \to Z_2}}, S_{x_2}), \quad x_{Z_1 \to Z_2 \to Z_1} = De(t_2). \tag{5}$$

Now, the same cycle process is again applied to transfer the speaking style of $Z_2$ to $Z_1$, we get following equations:

$$S_{y_1} = En(y_1), \quad S_{x_1} = En(x_1), \quad t'_1 = AdaIN(S_{y_1}, S_{x_1}), \quad y_{Z_2 \rightarrow Z_1} = De(t'_1), \quad (6)$$

$$S_{y_2} = En(y_2), \quad S_{y_{Z_2 \rightarrow Z_1}} = E(y_{Z_2 \rightarrow Z_1}), \quad t'_2 = AdaIN(S_{y_{Z_2 \rightarrow Z_1}}, S_{y_2}), \quad y_{Z_2 \rightarrow Z_1 \rightarrow Z_2} = De(t'_2).$$
$$(7)$$

During testing, we gave features of the source speaker's speech along with the sample features of target speaker to the encoder. AdaGAN requires 3 s to 5 s of sample speech features of the target speaker in order to transfer speaking style of target speaker to source speaker. This sample speech will be used to estimate the mean and standard deviation of the target speaker's distribution in its respective latent space. After this, the speaking style will be transferred in latent space of source speaker using $AdaIN$. Next, the decoder will generate speech back from the converted latent representation of the source speaker. Briefly, the decoder will generate the speech with speaking style of the target speaker. Now, the generator of AdaGAN is consist of Encoder and Decoder. Hence, we can say that the generator of AdaGAN will generate the speech with speaking style of the target speaker for a given source speaker's speech along with the sample of target speaker during testing. The training procedure of AdaGAN is formally presented in Algorithm 1.

---

**Algorithm 1** Algorithm for training of AdaGAN

---

**Input:** Weights of Encoder, Decoder, and Discriminator
**Output:** Optimized weights
1: **for** number of training iterations **do**
2:     randomly select two speakers ($Z_1$ and $Z_2$)
3:     sample 4 minibatches of cepstral features $\{x_1, x_2\} \in p_X(.|Z_1, U_1)$, and $\{y_1, y_2\} \in p_Y(.|Z_2, U_2)$.
4:
5:     /* Comment starts:
6:     First column shows the process of transferring speaking style of speaker $Z_2$ to $Z_1$.
7:     Second column shows the process of transferring speaking style of speaker $Z_1$ to $Z_2$.
8:     Comment ends */
9:
10:     $S_{X_1} \leftarrow En(x_1);$           $S_{Y_1} \leftarrow En(y_1);$
11:     $t_1 \leftarrow AdaIN(S_{X_1}, S_{Y_1});$     $t'_1 \leftarrow AdaIN(S_{Y_1}, S_{X_1});$
12:     $x' \leftarrow De(t_1);$             $y' \leftarrow De(t'_1);$
13:     $S_{X_2} \leftarrow En(x_2);$           $S_{Y_2} \leftarrow En(y_2);$
14:     $S_{X'} \leftarrow En(x');$           $S_{Y'} \leftarrow En(y');$
15:     $t_2 \leftarrow AdaIN(S_{X'}, S_{Y_2});$     $t'_2 \leftarrow AdaIN(S_{Y'}, S_{X_2});$
16:
17:     Update the generator by descending its stochastic gradient:

    $\theta_{En,De} \overset{+}{\leftarrow} \delta_{\theta_{En,De}}(\mathcal{L}_{adv} + \lambda_1 \mathcal{L}_{cyc} + \lambda_2 \mathcal{L}_{C_{X \rightarrow Y}} + \lambda_3 \mathcal{L}_{C_{Y \rightarrow X}} + \lambda_4 \mathcal{L}_{sty_{X \rightarrow Y}} + \lambda_5 \mathcal{L}_{sty_{Y \rightarrow X}})$

18:
19:     Update the discriminator by descending its stochastic gradient:

    $\theta_{Dis} \overset{+}{\leftarrow} \delta_{\theta_{Dis}}(\mathcal{L}_{adv})$

20:
21: **end for**
22: **return**

---

### 4.3 LOSS FUNCTIONS

To achieve many-to-many and zero-shot VC, AdaGAN uses four different loss functions: Adversarial loss, reconstruction loss, content preserve loss, and style transfer loss.

**Adversarial loss:** This loss measures how distinguishable the converted data is from the normal speech data. The smaller the loss is, the converted data distribution is more closer to normal speech distribution. Hence, we want to minimize objective function given in eq. (9) against an adversary Dis(.) that tries to maximize it. Here, this loss is used to make the generated or converted speech

indistinguishable from the original speech, and can be mathematically formulated as:

$$\mathcal{L}_{adv}(En, De) = (Dis(y_{Z_2 \to Z_1}) - 1)^2 + (Dis(x_{Z_1 \to Z_2}) - 1)^2, \tag{8}$$

$$\mathcal{L}_{adv}(Dis) = (Dis(x_1) - 1)^2 + (Dis(y_1) - 1)^2. \tag{9}$$

**Reconstruction Loss:** By using only adversarial loss, we may loose linguistic information in the converted voice. This loss helps the encoder and decoder to retain the linguistic information in converted voice. We have used $L_1$ norm as a reconstruction loss, and can be described as:

$$\mathcal{L}_{cyc} = \|x_{Z_1 \to Z_2 \to Z_1} - x_1\|_1 + \|y_{Z_2 \to Z_1 \to Z_2} - y_1\|_1. \tag{10}$$

**Content Preserve Loss:** To preserve the linguistic content of the input speech during *AdaIN*. This loss also ensure that our encoder and decoder are noise free. We have used following $L_1$ norm for this loss, i.e.,

$$\mathcal{L}_{C_{X \to Y}} = \|S_{x_{Z_1 \to Z_2}} - t_1\|_1. \tag{11}$$

**Style transfer Loss:** This loss function is at the heart of the AdaGAN. This loss plays a vital role in achieving many-to-many and zero-shot VC using AdaGAN. This loss helps AdaGAN to create a latent space with the speaking style features in terms of mean and standard deviation of the distribution while preserving the linguistic content in the same distribution. We have used $L_1$ norm as style transfer loss, i.e.,

$$\mathcal{L}_{sty_{X \to Y}} = \|t_2 - S_{X_1}\|_1, \tag{12}$$

**Final Objective Function:** The overall objective function of AdaGAN can be defined as:

$$\begin{aligned} \mathcal{L}_{total} =& \mathcal{L}_{adv}(En, De) + \mathcal{L}_{adv}(Dis) + \lambda_1 \mathcal{L}_{cyc} + \lambda_2 \mathcal{L}_{C_{X \to Y}} + \\ & \lambda_3 \mathcal{L}_{C_{Y \to X}} + \lambda_4 \mathcal{L}_{sty_{X \to Y}} + \lambda_5 \mathcal{L}_{sty_{Y \to X}}, \end{aligned} \tag{13}$$

where $\lambda_1$, $\lambda_2$, $\lambda_3$, $\lambda_4$, and $\lambda_5$ are the hyperparameters. These parameters controls the relative importance of each loss w.r.t. each other. We have used $\lambda_1 = 10$, $\lambda_2 = 2$, $\lambda_3 = 2$, $\lambda_4 = 3$, and $\lambda_5 = 3$ during the experiments. We theoretically proved that how these simple loss functions are the key idea behind the performance of AdaGAN in the next Section. We optimized these loss functions according to the Algorithm 1.

## 4.4 ARCHITECTURAL DETAILS

AdaGAN framework contains a Generator and a Discriminator. In this Section, we provide detailed information about each component of the AdaGAN framework.

As shown in Fig. 1, Generator of AdaGAN consists of mainly 2 modules: Encoder and Decoder. AdaGAN uses the same encoder to extract the features from the source and target speakers' speech. Input of encoder is a vector of 40 Mel cepstral features, which it converts to a latent space of size 1x512. The decoder takes normalized feature vector of size 1x512 as input and converts it to 1x40 target speech features.

In encoder and decoder, all layers are fully-connected layers. In encoder, the input and output layer has 40 and 512 cell size, respectively. In decoder, input and output layer have 512 and 40 cell size, respectively. All the hidden layers in encoder and decoder consist 512 cell size. All the layers are followed by Rectified Linear Unit (ReLU) activation function except output layer.

In AdaGAN, main goal of the discriminator is similar to traditional GAN training. Accurately, it will discriminate whether the input is generated ($x_{Z_1 \to Z_2}$) or from the original distribution. Same as Encoder and Decoder, structure of discriminator follows the stacked fully-connected layers. It consists of an input layer, 3 hidden layers and, an output layer with 40, 512, and 1 cell size, respectively. In discriminator, each layer followed by the ReLU activation function and output layer followed by a sigmoid activation function.

## 5 ANALYSIS AND COMPARISON

In this Section, we show the theoretical correctness and intuitive explanation of AdaGAN. The key idea of the AdaGAN is to learn the latent space, where we can represent our features as per our requirements.

### 5.1 THEORETICAL ANALYSIS

Consider the training procedure of AdaGAN described in Section 4.2. Let us take two latent space features $S_{x_1}$ and $S_{x_2}$ corresponding to two different sample features, $x_1$ and $x_2$, respectively, of the same speaker $Z_1$. We are also going to take $S_{y_1}$ from latent space of another speaker $Z_2$, where $y_1$ is a sample feature of that speaker, and $Z_1 \neq Z_2$. After training of AdaGAN for a large number of iteration of $\tau$, where theoretically $\tau \to \infty$, let us assume the following:

1. In the latent space, mean and standard deviation of the same speaker are constant irrespective of the linguistic content. Formally, we have $\mu_{S_{x_1}} = \mu_{S_{x_2}}$, and $\sigma_{S_{x_1}} = \sigma_{S_{x_2}}$.

2. If we have different speakers, then mean and standard deviation of respective latent representations are different. Accurately, $\mu_{S_{x_1}} \neq \mu_{S_{y_1}}$, and $\sigma_{S_{x_1}} \neq \sigma_{S_{y_1}}$.

**Theorem 1:** Given these assumptions, $\exists$ a latent space where normalized latent representation of input features will be the same irrespective of speaking style. Here, we take input features of same utterance $U_1$. Hence,

$$D_{\text{KL}}( \quad p_{IN}(.|Z_1, U_1) \quad \| \quad p_{IN}(.|Z_2, U_1) \quad ) = 0, \tag{14}$$

where $KL(\cdot|\cdot)$ is the KL-divergence, and $p_N(.|Z_i, U_i)$ is *pdf* of normalized latent representation of input feature $U_i$, with speaking style of speaker $Z_i$.

This is the fundamental theorem that lies behind the concept of AdaGAN. Intuitively, from this theorem, we can observe that the normalized latent representation of the same utterance spoken by different speakers is the same. This fact leads to the conclusion that linguistic content of speech is captured by the distribution of normalized latent space, and speaking style of a speaker is being captured by mean and standard deviation of the same distribution.

**Theorem 2:** By optimization of $\min_{En, De} \mathcal{L}_{C_{X \to Y}} + \mathcal{L}_{sty_{X \to Y}}$, the assumptions made in Theorem 1 can be satisfied.

The proof of both the theorems are given in Appendix A. Both the theorems conclude that AdaIN made style transfer easy and efficient via only using the mean and standard deviation of the distribution. In Appendix B, we provided the t-SNE visualization of the features in latent space to give the empirical proof.

### 5.2 ADAGAN *vs.* STARGAN

In this Section, we show a comparison between AdaGAN and StarGAN-VC in terms of computational complexity. Table 1 and 2 provided the number of layers, FLoating point Operations Per Second (FLOPS), and trainable parameters[1] for the AdaGAN and StarGAN-VC, respectively.

Table 1: Number of layers and trainable parameters in AdaGAN

| Module | Layers | FLOPS | Trainable Parameters |
|---|---|---|---|
| $G_{AdaGAN}$ | 8 fully-connected layers | 4,271,576,064 | 2,142,760 |
| $Dis_{AdaGAN}$ | 4 fully-connected layers | 1,612,798,976 | 809,473 |
| | Total FLOPS and parameters: | 5,884,375,040 | 2,952,233 |

---

[1] All the parameters are calculated using the *thop* library, version - $0.0.31.post1909021322$.

Table 2: Number of layers and trainable parameters in StarGAN

| Module | Layers | FLOPS | Trainable Parameters |
|---|---|---|---|
| $G_{StarGAN}$ | 18 conv layer | 15,751,839,744 | 9,073,536 |
| $D_{StarGAN}$ | 6 conv layer | 35,858,677,760 | 1,152,320 |
| $Cls$ | 6 conv layer | 51,904,512 | 81,920 |
| | Total FLOPS and parameters: | 51,662,422,016 | 20,307,776 |

In Table 1, $G_{AdaGAN}$ and $D_{AdaGAN}$ are the generator and discriminator of AdaGAN, respectively. Parameters of the generator are calculated by adding the parameters of encoder and decoder. To calculate the FLOPS and parameters for StarGAN, we have used the open-source implementation of StarGAN-VC[2]. In the Table 2, $G_{StarGAN}$, $D_{StarGAN}$, and $Cls$ are generator, discriminator, and classifier of StarGAN, respectively. All these three modules contain convolution layers. In StarGAN, there is weight sharing between the 5 convolution layers of discriminator and classifier. Here, we remove the FLOPS and trainable parameters of shared layers from the $Cls$. Hence, we consider it once in the calculation of total FLOPS and trainable parameters.

We can observe that AdaGAN is 88.6% less complex than StarGAN in terms of FLOPS, and 85.46% less complex in terms of trainable parameters. Moreover, StarGAN uses a one-hot encoding to get the information about the target speaker. However, AdaGAN requires any sample of 3 s - 5 s from the target speaker.

## 6    EXPERIMENTAL RESULTS

In this Section, we will show experimental setup, and subjective evaluation (or results) of AdaGAN. Samples of converted audio files are provided here[3].

### 6.1    EXPERIMENTAL SETUP

The experiments are performed on the VCTK corpus (Veaux et al. (2017)), which contains 44 hours of data for 109 speakers. The statistics of the database are given Veaux et al. (2017). The database is designed to provide non-parallel data for VC. From this database, AdaGAN system was developed on data of 20 speakers (10 males and 10 females). Out of this, we have used 80% data for training and 20% data for testing for each speaker. Particularly, we have used $6.27$ and $1.45$ hours of data for the training and testing, respectively. The *40*-dimensional (dim) Mel Cepstral Coefficients (MCCs) (including the $0^{th}$ coefficient) and *1*-dimensional $F_0$ are extracted from the speech of source, and the target speakers with 25 ms window and 5 ms frame-shift. For analysis-synthesis, we have used AHOCODER (Erro et al. (2011)). Mean-variance transformation method has been applied for fundamental frequency $F_0$ conversion Toda et al. (2007).

To evaluate AdaGAN empirically, we performed two subjective tests for evaluating naturalness and speaker similarity. In particular, Mean Opinion Score (MOS) test have been conducted, where subjects have been asked to rate the randomly played converted speech on 5-point scale for naturalness, where 1 means converted voice is very robotic, and 5 means converted voice is very natural. In the second test, subjects have been asked to rate how similar the converted voice given the reference target speech in terms of speaker similarity. Subjects rated converted voices for speaker similarity on the 5-point scale, where 1 means dissimilar with high confidence and 5 means similar with high confidence w.r.t. the given target speaker. Total 15 subjects (6 females and 9 males with no known hearing impairments with age varies between 18 to 31 years) took part in the subjective evaluations.

### 6.2    RESULTS FOR MANY-TO-MANY CONVERSION

Randomly 2 male and 2 female speakers have been selected from the testing dataset for subjective evaluations. We evaluated four different conversion systems, i.e., male-male (M2M), female-female

---

[2]`https://github.com/liusongxiang/StarGAN-Voice-Conversion`
[3]`https://drive.google.com/open?id=1VzA2bRhUz1lZ4DBDOIKUO9wOM8xkcTj2`

(F2F), male-female (M2F), and female-male (F2M) developed using proposed AdaGAN and Star-GAN. From each system, two converted audio files have been selected. Hence, 8 audio files from AdaGAN and another 8 audio files from the StarGAN have been taken for subjective evaluations. We kept the same source-target speaker-pairs for fair comparison.

Fig. 2 shows the comparison of MOS scores between AdaGAN and the baseline StarGAN-VC. Total of 15 subjects (6 females and 9 males) between 18-30 years of age and with no known hearing impairments took part in the subjective test. For statistically significant analysis, results are shown in different conversion possibilities with 95% confidence interval. In addition, for our subjective tests, we obtain p-value 0.013, which is much lesser then 0.05. Therefore, it clearly shows the statistical significance of the results. From Fig. 2, it is clear that there is 31.73 % relative improvement (on an average) in MOS score for the AdaGAN compared to the baseline StarGAN. In terms of speaker similarity, AdaGAN yields on an average 10.37% relative improvement in speaker similarity compare to baseline (as shown in Fig. 3). Although AdaGAN outperforms StarGAN, both the methods are not able to achieve good score in the similarity test. The main reason is due to the $F_0$ conversion and errors in statistical vocoder (i.e., AHOCODER and WORLD-vocoder). However, neural network-based Wavenet-vocoder shows very promising results on speech synthesis. Although they are very accurate, they are data-driven approaches. In summary, AdaGAN achieves better performance in MOS tests compared to the StarGAN-VC for naturalness and speaker similarity.

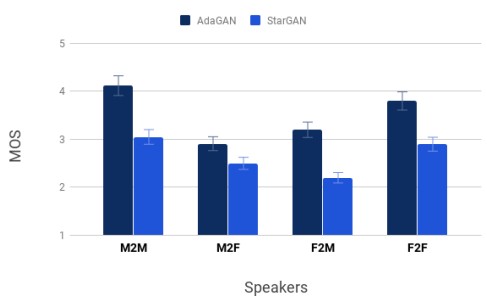

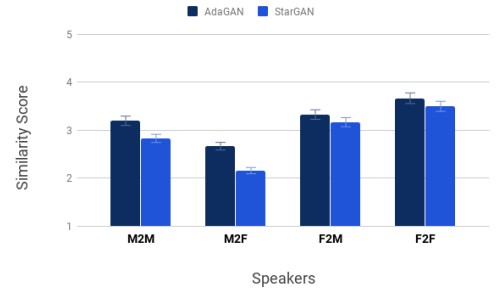

Figure 2: MOS for naturalness of AdaGAN and StarGAN with 95% confidence interval.

Figure 3: MOS for speaker similarity of Ada-GAN and StarGAN with 95% confidence interval.

## 6.3 ZERO-SHOT LEARNING

In traditional many-to-many VC, all the target speakers are seen while training the architecture. Hence, traditional algorithms are not able to do VC for an unseen speaker (i.e., for the cases of zero-shot VC). Along with many-to-many VC, we extended our study of AdaGAN for zero-shot VC. Zero-shot conversion is the task of transferring the speaking style of seen/unseen source speaker to seen/unseen target speaker. In simple terms, conversion can be done between any speaker whether their data were present in the corpus or not at the time of training. StarGAN-VC uses a one-hot vector for target speaker reference during conversion. In the case of an unseen target speaker, it will not be able to perform the zero-shot conversion. However, AdaGAN maps the input to the required latent space (as proved in Appendix A). Therefore, AdaGAN will be able to learn more promised latent space for even unseen speakers. Here, we show our experimental results for the zero-shot VC task. We performed subjective tests in a similar manner as performed in many-to-many VC. We have used AdaGAN trained on 20 speakers (10 males and 10 females). Later on, we selected randomly 1 seen, and 1 unseen male speakers and 1 seen, and 1 unseen female speakers. And we applied the permutations on these different speakers to get all the different conversion samples, such as seen-seen (S2S), seen-unseen (S2U), unseen-seen (U2S), and unseen-unseen (U2U). Fig. 4, and Fig. 5 shows the MOS scores for naturalness, and speaker similarity, respectively.

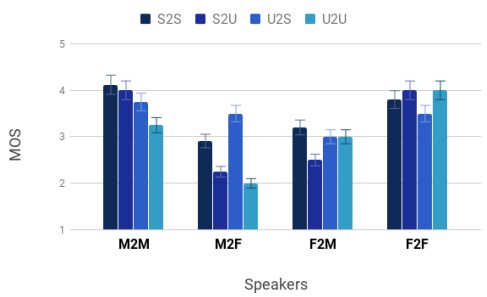
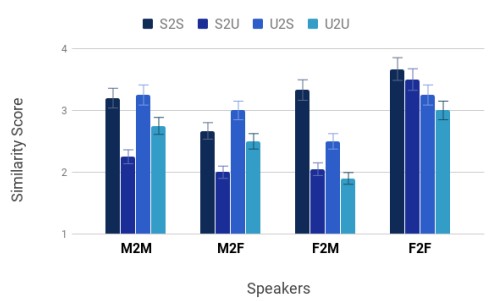

Figure 4: MOS for naturalness of AdaGAN for zero-shot VC with 95% confidence interval.

Figure 5: MOS for Speaker Similarity of Ada-GAN for zero-shot VC with 95% confidence interval.

Recently, AutoVC has been proposed, which is the only framework for zero-shot conversion VC Qian et al. (2019). To the best of authors' knowledge, this is the first GAN-based framework to achieve zero-shot VC. To do the zero-shot conversion, AutoVC requires few samples (20 s) of possible target speakers. However, AdaGAN requires only 3s to 5s of sample speech of the seen or unseen target speaker to extract latent representation for the target speaker in order to generate voices that sound perceptually similar to the target speaker. Moreover, trained AdaGAN architecture can work on any source or target speaker.

## 7 CONCLUSIONS AND FUTURE WORK

In this paper, we proposed novel AdaGAN primarily for non-parallel many-to-many VC task. Moreover, we analyzed our proposed architecture w.r.t. current GAN-based state-of-the-art StarGAN-VC method for the same task. We know that the main aim of VC is to convert the source speaker's voice into the target speaker's voice while preserving linguistic content. To achieve this, we have used the style transfer algorithm along with the adversarial training. AdaGAN transfers the style of the target speaker into the voice of a source speaker without using any feature-based mapping between the linguistic content of the source speaker's speech. For this task, AdaGAN uses only one generator and one discriminator, which leads to less complexity. AdaGAN is almost 88.6% computationally less complex than the StarGAN-VC. We have performed subjective analysis on the VCTK corpus to show the efficiency of the proposed method. We can clearly see that AdaGAN gives superior results in the subjective evaluations compared to StarGAN-VC.

Motivated by the work of AutoVC, we also extended the concept of AdaGAN for the zero-shot conversion as an independent study and reported results. AdaGAN is the first GAN-based framework for zero-shot VC. In the future, we plan to explore high-quality vocoders, namely, WaveNet, for further improvement in voice quality. The perceptual difference observed between the estimated and the ground truth indicates the need for exploring better objective function that can perceptually optimize the network parameters of GAN-based architectures, which also forms our immediate future work.

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

# A   MATHEMATICAL PROOF OF ADAGAN CONCEPT AND DIFFERENT LOSS FUNCTIONS

Here, we will first give the proof of the concept behind AdaGAN and later we will prove how our loss functions help us to satisfy the derived constrained in Theorem 1.

**Theorem 1:** Given assumptions in Section 5.1, we can say that there exists a latent space where normalized latent representation of input features will be the same irrespective of speaking style.

**Proof:** From eq. (1), we can write the goal of AdaGAN as following:

$$D_{\mathrm{KL}}\big(\quad p_{\hat{X}_{Z_1 \to Z_2}}(.|Z_1, U_1, Z_2, U_2) \quad \| \quad p_{\hat{X}_{Z_2 \to Z_2}}(.|Z_2, U_1, Z_2, U_2') \quad \big) = 0. \tag{15}$$

From eq. (15), we can conclude that the output of AdaGAN after conversion using $(Z_1, U_1)$ and $(Z_2, U_2)$ is the same as conversion using $(Z_1, U_1)$ and $(Z_2, U_2)$. Because in either way, we are transferring speaking style of speaker $Z_2$ to utterance $U_1$. We can say that the output for $AdaIN$ is the same for both the cases in the latent space. Hence, we can write above eq. (15) as:

$$\implies D_{\mathrm{KL}}\big(\quad p_{AdaIN}(.|Z_1, U_1, Z_2, U_2) \quad \| \quad p_{AdaIN}(.|Z_2, U_1, Z_2, U_2') \quad \big) = 0, \tag{16}$$

where $p_{AdaIN}(.|.)$ is the *pdf* of the latent representation. For given input samples $x_1$ and $y_1$, we can write following term from eq. (16):

$$p_{AdaIN_{x_1}}(.|Z_1, U_1, Z_2, U_2) \quad = \quad p_{AdaIN_{y_1}}(.|Z_2, U_1, Z_2, U_2'), \tag{17}$$

From Fig. 1, we can write eq. (17) as:

$$\implies \left[\frac{S_{x_1}(\tau) - \mu_1(\tau)}{\sigma_1(\tau)}\right]\sigma_2(\tau) + \mu_2(\tau) \quad = \quad \left[\frac{S_{y_1}(\tau) - \mu_2''(\tau)}{\sigma_2''(\tau)}\right]\sigma_2'(\tau) + \mu_2'(\tau),$$

where $\tau$ represents the training iteration. Now, giving $\lim_{\tau \to \infty}$ both side we assume that $\mu_2''(\tau) = \mu_2'(\tau) = \mu_2(\tau)$, and $\sigma_2''(\tau) = \sigma_2'(\tau) = \sigma_2(\tau)$. Therefore,

$$\left[\frac{S_{x_1}(\tau) - \mu_1(\tau)}{\sigma_1(\tau)}\right]\sigma_2(\tau) + \mu_2(\tau) \quad = \quad \left[\frac{S_{y_1}(\tau) - \cancel{\mu_2''(\tau)}}{\cancel{\sigma_2''(\tau)}}\right]\cancel{\sigma_2'(\tau)} + \cancel{\mu_2'(\tau)},$$

$$\implies \left[\frac{S_{x_1}(\tau) - \mu_1(\tau)}{\sigma_1(\tau)}\right] \quad = \quad \left[\frac{S_{y_1}(\tau) - \mu_2(\tau)}{\sigma_2(\tau)}\right]. \tag{18}$$

At $\tau \to \infty$, the assumptions that made in Section 5.1 are true. Hence, from eq. (18), we can conclude that there exists a latent space where normalized latent representation of input features will be the same irrespective of speaking style.

**Theorem 2:** By optimization of $\min_{En,De} \mathcal{L}_{C_{X \to Y}} + \mathcal{L}_{sty_{X \to Y}}$, the assumptions made in Theorem 1 can be satisfied.

**Proof:** Our objective function is the following:

$$\min_{En,De} \mathcal{L}_{C_{X \to Y}} + \mathcal{L}_{sty_{X \to Y}}. \tag{19}$$

Iterate step by step to calculate the term $(t_2)$ used in loss function $\mathcal{L}_{sty_{X \to Y}}$. Consider, we have the latent representations $S_{x_1}$ and $S_{y_1}$ corresponding to the source and target speech, respectively.

$$\text{Step 1:} \quad \left[\frac{S_{x_1}(\tau) - \mu_1(\tau)}{\sigma_1(\tau)}\right]\sigma_2(\tau) + \mu_2(\tau) \quad \text{(Representation of } t_1\text{)},$$

$$\text{Step 2\&3:} \quad En\left\{De\left[\left[\frac{S_{x_1}(\tau) - \mu_1(\tau)}{\sigma_1(\tau)}\right]\sigma_2(\tau) + \mu_2(\tau)\right]\right\}.$$

After applying decoder and encoder sequentially on latent representation, we will again get back to the same representation. This is ensured by the loss function $\mathcal{L}_{C_{X \to Y}}$. Formally, we want to make $\mathcal{L}_{C_{X \to Y}} \to 0$. Therefore, we can write step 4 as:

Step 4: $\qquad \left[\dfrac{S_{x_1}(\tau) - \mu_1(\tau)}{\sigma_1(\tau)}\right]\sigma_2(\tau) + \mu_2(\tau) \qquad$ (i.e., reconstructed $t_1$),

Step 5: $\qquad \dfrac{1}{\sigma_2(\tau)}\left[\left[\dfrac{S_{x_1}(\tau) - \mu_1(\tau)}{\sigma_1(\tau)}\right]\sigma_2(\tau) + \mu_2(\tau) - \mu_2(\tau)\right]$

$\qquad$ (Normalization with its own (i.e., latent representation in Step 4) $\mu$ and $\sigma$ during $AdaIN$),

Step 6: $\qquad \left[\dfrac{S_{x_1}(\tau) - \mu_1(\tau)}{\sigma_1(\tau)}\right] \qquad$ (Final output of Step 5),

Step 7: $\qquad \left[\dfrac{S_{x_1}(\tau) - \mu_1(\tau)}{\sigma_1(\tau)}\right]\sigma_1'(\tau) + \mu_1'(\tau)$

$\qquad$ (Output after de-normalization in $AdaIN$. Representation of $t_2$),

where $\mu_1'$ and $\sigma_1'$ are the mean and standard deviations of the another input source speech, $x_2$. Now, using the mathematical representation of $t_2$, we can write loss function $\mathcal{L}_{sty_{X \to Y}}$ as:

$$\mathcal{L}_{sty_{X \to Y}} = \left[\left(\dfrac{S_{x_1}(\tau) - \mu_1(\tau)}{\sigma_1(\tau)}\right)\sigma_1'(\tau) + \mu_1'(\tau) - S_{x_1}(\tau)\right]. \tag{20}$$

According to eq. (19), we want to minimize the loss function $\mathcal{L}_{sty_{X \to Y}}$. Formally, $\mathcal{L}_{sty_{X \to Y}} \to 0$. Therefore, we will get $\mu_1 = \mu_1'$, and $\sigma_1 = \sigma_1'$ to achieve our goal. Hence, mean and standard deviation of the same speaker are constant, and different for different speakers irrespective of the linguistic content. We come to the conclusion that our loss function satisfies the necessary constraints (assumptions) required in proof of Theorem 1.

## B   T-SNE VISUALIZATION OF LATENT SPACE LEARNED BY ADAGAN

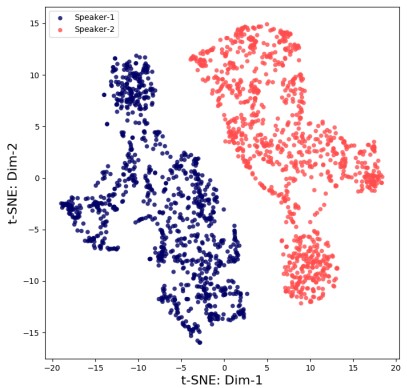
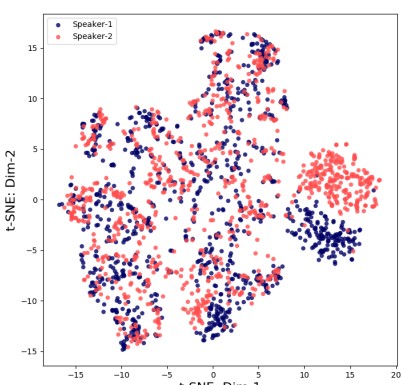

Figure 6: t-SNE visualization of latent representation of two speakers' speech and its normalized form, where, each point denotes a feature extracted from the 25 ms of speech segment.

As we know, Neural Networks (NNs) are hard to train and optimize. Even if everything has been proven in terms of theoretical proofs, statistical and empirical analysis is required. For this analysis, we have adopted t-SNE visualization. Here, we randomly selected few utterances from two different speakers from the VCTK corpus. Latent representations are extracted for the speech of that speakers, and features are reduced to 2-D using t-SNE. The scatter plot shown in Fig. 6 shows that data points are clustered based on the speaking style. After normalized with their respective means and standard deviations, these distribution overlapped. This shows that the distribution of normalized latent representation captures linguistic information-based features irrespective of speaking style as proved in Theorem 1. Therefore, we can say that AdaGAN, and its losses are efficient for practical purposes.

