# OpenReview forum: "AdaGAN: Adaptive GAN for Many-to-Many Non-Parallel Voice Conversion"
_ICLR.cc/2020/Conference — Reject_

### Official Review · AnonReviewer2 · 2019-10-23
**Official Blind Review #2**

**Rating:** 1

**Review:**

This paper presents a voice conversion approach using GANs based on adaptive instance normalization (AdaIN).  The authors give the mathematical formulation of the problem and provide the implementation of the so-called AdaGAN. Experiments are carried out on VCTK and the proposed AdaGAN is compared with StarGAN.  The idea is ok and the concept of using AdaIN for efficient voice conversion is also good.  But the paper has a lot of issues both technically and grammatically, which makes the paper hard to follow.

1. On writing
There are glaring grammar errors in numerous places. e.g.
  -- "Although, there are few GAN-based systems that produced state-of-the-art results for non-parallel VC. Among these algorithms, even fewer can be applied for many-to-many VC task. At last, there is the only system available for zero-shot VC proposed by Qian et al. (2019)."   This is hard to parse.
 -- "helps generator to make ..."  -> "helps the generator make ..."
 --  "let assume" -> "Let's assume"
 --  "We know that the idea of transitivity as a way to regularize structured data has a long history."   what does it mean?
 --  "the generator of AdaGAN is consists of Encoder and Decoder."  -> "consist of"
 -- "After training of AdaGAN for large number of iteration of $\tau$ , where theoretically $\tau \rightarrow \infty$." where is the second half of the sentence?

2.  On math notation
 The math notation is messy and there are lots of inaccuracies.  e.g.
  --  $X_{i} \in p_{X}(\cdot|Z_{i},U_{i})$ should be $X_{i} \sim p_{X}(\cdot|Z_{i},U_{i})$
  --  "generate the distribution denoted by $\hat{X}_{Z_{1}\rightarrow Z_{2}}$"  -> why  $\hat{X}_{Z_{1}\rightarrow Z_{2}}$ becomes a distribution?
  --  "$p_{N}(\cdot|Z_{1},U_{1})$, $p_{N}(\cdot|Z_{2},U_{1})$" in Eq.14,  $N$ should be replaced by the random variable.
  --  $S'_{X}$ and $S'_{Y}$ should be $S_{X'}$ and $S_{Y'}$ in line 15 in the algorithm

3. On technical details:
 -- In Fig.1 (b), why is there only one input to the discriminator?  How do you inject the adversarial samples and how do you generate adversarial samples?
-- In section 4.4, "in encoder and decoder all layers are Linear layers".  Are you referring to fully-connected layers? Linear layers are usually referred to those with linear activation functions.
-- The experiments are claimed to be zero-shot, but 3-5s of speech is required.  can you explain?

Although the samples sound OK, given its current form, the paper needs significant re-work.

P.S. rebuttal read.   I will stay with my score.

**Experience Assessment:**

I have read many papers in this area.

**Review Assessment: Checking Correctness Of Derivations And Theory:**

I assessed the sensibility of the derivations and theory.

**Review Assessment: Checking Correctness Of Experiments:**

I assessed the sensibility of the experiments.

**Review Assessment: Thoroughness In Paper Reading:**

I read the paper at least twice and used my best judgement in assessing the paper.

---

> ### Author Response · Authors · 2019-11-14
> **Response to Reviewer #2**
>
> We sincerely thank respected Reviewer 2 for these valuable comments. These comments were highly useful to improve the updated manuscript. We sincerely believe that the reviewer’s concerns addressed additional clarifications in the updated version of the paper. We hope the reviewer will agree. We provide further details below in this regard.
>
> Q 1. On writing There are glaring grammar errors in numerous places. e.g.
> -- "Although, there are few GAN-based systems that produced state-of-the-art results for non-parallel VC. Among these algorithms, even fewer can be applied for many-to-many VC tasks. At last, there is the only system available for zero-shot VC proposed by Qian et al. (2019)." This is hard to parse.
>
> Ans: Thanks for these suggestions. We have gone through several rounds of reading to take care of the grammatical errors. From the above paragraph, we are trying to convey the message that, “Among all the methods which exist for many-to-many VC, only a few can work on non-parallel data. Among these non-parallel many-to-many VC approaches, there is only one method proposed by Qian et al. (2019), which work for the zero-shot VC task as well [1].”
>
> Q 2. "After training of AdaGAN for large number of iterations, where theoretically ." where is the second half of the sentence?
>
> Ans: Thank you very much for pointing out our mistakes, we are very sorry for this. Actually, by mistake, a full stop was added instead of commas. We have updated this mistake in section 5.1 and also gone through the whole paper to make it grammatically correct. Many Thanks!
>
> Q 3. "We know that the idea of transitivity as a way to regularize structured data has a long history." what does it mean?
>
> Ans: It was written in different context. We have now removed this sentence. We are very sorry for this.
>
> Q 4. Math notation issue: $X_i \in p_X(.|Z_i,U_i)$ should be $X_i \sim p_X(.|Z_i,U_i)$
>
> Ans: Yes, it was a mistake from our side. Thank you for bringing out this key issue. We have corrected it.
>
> Q 5. Math notation issue: "generate the distribution denoted by $\hat{X}_{Z_1\rightarrow Z_2}$" -> why $\hat{X}_{Z_1\rightarrow Z_2}$ becomes a distribution?
>
> Ans: $\hat{X}_{Z_1\rightarrow Z_2}$ is denoting a distribution change from $p_{X}(.|Z_1, U_1)$ to $p_{X}(.|Z_2, U_1)$. This indicates that $\hat{X}_{Z_1\rightarrow Z_2}$ is distribution containing linguistic information $U_1$ and speaking style of $Z_2$.
>
> Q 6. Math notation issue: "$p_{N}(.|Z_1, U_1)$, $p_{N}(.|Z_2, U_1)$" in Eq.14, $N$ should be replaced by the random variable.
>
> Ans: To avoid confusion, now, we have changed the notation to: $p_{IN}(.|Z_1, U_1)$, $p_{IN}(.|Z_2, U_1)$ (in section 5.1, eq. (14)). Here, we use IN to denote that respective distributions are after the normalization during AdaIN step.
>
> Q 7. Math notation issue: $S_X'$ and $S_Y'$ should be $S_{X'}$ and $S_{Y'}$ in line 15 in the algorithm.
>
> Ans: Yes, instead of $S_X'$ and $S_Y'$ there should be $S_{X'}$ and $S_{Y'}$. We are sorry for this typo. We have made changes. Thanks this suggestion!
>
> Q 8. On technical details: -- In Fig.1 (b), why is there only one input to the discriminator? How do you inject the adversarial samples and how do you generate adversarial samples?
>
> Ans: We have shown our adversarial training in Eq. (8) and Eq. (9), in that we have clearly shown that discriminator takes both the input generated features as well as original features. To show our training process simple and to make the figure straightforward, we did not indicate original features as input to the discriminator. However, we have made this change in Fig. (1) in the revised manuscript.
>
> Q 9. In section 4.4, "in encoder and decoder all layers are Linear layers". Are you referring to fully-connected layers? Linear layers are usually referred to those with linear activation functions.
>
> Ans: Yes, we are referring to fully-connected layers. Thanking you for drawing our attention to this point. We have now updated this in section 4.4 of revised manuscript.
>
> Q 10. The experiments are claimed to be zero-shot, but 3-5s of speech is required. can you explain?
>
> Ans: Yes, we claim experiments to be zero-shot. We do understand that zero-shot means that we do not require any target speakers' data during training procedure. Here, AdaGAN requires 3-5 seconds from target speakers’ speech during the testing phase only and just as a reference to the target. The carefully designed training procedures allow AdaGAN to generate the latent representation of unknown speakers' speech via some reference utterance of the same speaker. The AutoVC paper which proposed the first attempt to zero-shot VC via encoders and decoders, which requires 20 seconds of target speakers' speech as a reference [1].
>
>
>
> [1] Kaizhi Qian, Yang Zhang, Shiyu Chang, Xuesong Yang, and Mark Hasegawa-Johnson. Autovc: Zero-shot voice style transfer with only autoencoder loss. In International Conference on Machine Learning (ICML), pp. 5210–5219, 2019.

---

### Official Review · AnonReviewer3 · 2019-10-23
**Official Blind Review #3**

**Rating:** 1

**Review:**

This paper tackles many-to-many voice conversion task using GAN for style transfer between different speakers. The core idea is adaptive instance normalization (Huang & Belongie, 2017).

Detailed comments:

There are many typos and wrong notations in the text. Here is an incomplete list:
- "it were spoken by target speaker", should be "was".
- In Section 3.1, "Here, U_1 and U_2 are spoken by Z_i and Z_2" should Z_1. Overall, the descriptions in this subsection is confusing. For example, it seems utterance U_i is from speaker Z_i in the dataset, but there are n speakers and m utterances.

- A closely related task is voice cloning, which is arguably more challenging than voice conversion, because the synthesis need generalize to arbitrary text. One may properly discuss the recent advances in this community (e.g., Arik et al., 2018; Nachmani et al., 2018; Jia et al., 2018).

Pros:
The empirical improvement seems meaningful.

Cons:
This paper is poorly written and difficult to follow. For example, I could not accurately identify the major contribution & novelty after reading the abstract and introduction. As an application paper, the authors may clearly explain the ideas with a few sentences in the most natural way without "heavy notations", e.g., Eq. (5)(6)(7).

**Experience Assessment:**

I have published in this field for several years.

**Review Assessment: Checking Correctness Of Derivations And Theory:**

I did not assess the derivations or theory.

**Review Assessment: Checking Correctness Of Experiments:**

I assessed the sensibility of the experiments.

**Review Assessment: Thoroughness In Paper Reading:**

I read the paper at least twice and used my best judgement in assessing the paper.

---

> ### Author Response · Authors · 2019-11-14
> **Response to Reviewer #3**
>
> Thank you for your valuable suggestions. We do believe that the reviewer’s concerns  are taken care through additional clarifications in the updated version of the paper. We hope the reviewer will agree. We provide further details below.
>
> Q 1. There are many typos and wrong notations in the text.
>
> Ans: Thank you for pointing out this issue. We have carefully reviewed our paper and made sure that there are no typos and wrong notation in the revised manuscript.
>
> Q 2. A closely related task is voice cloning, which is arguably more challenging than voice conversion, because the synthesis need generalize to arbitrary text. One may properly discuss the recent advances in this community (e.g., Arik et al., 2018; Nachmani et al., 2018; Jia et al., 2018).
>
> Ans: Yes, we do agree that voice cloning is more challenging. However, Voice Conversion has its problems that need to be solved. Through this study represented in our manuscript, we proposed a completely new way to approach Voice Conversion using latent representation learning. We have updated the related work according to recent advances in voice cloning, as well as per the reviewer’s suggestion. We sincerely thank respected reviewer.
>
> Q 3. This paper is poorly written and difficult to follow. For example, I could not accurately identify the major contribution & novelty after reading the abstract and introduction.
>
> Ans: We welcome this important feedback of language related corrections and we have taken serious note of it. We added a new paragraph at the end of the introduction section, which describes novelty and key contributions of the AdaGAN.
>
> Q 4. As an application paper, the authors may clearly explain the ideas with a few sentences in the most natural way without "heavy notations", e.g., Eq. (5)(6)(7).
>
> Ans: Yes, we do understand. However, we believe that by adding mathematical formulas, it will be helpful to future readers to implement and manage data processing for various applications of voice conversion.

---

### Official Review · AnonReviewer1 · 2019-10-28
**Official Blind Review #1**

**Rating:** 6

**Review:**

This work describes an efficient voice conversion system that can operate on non-parallel samples and convert from and to multiple voices.  The central element of the methodology is the AdaIn modification.  This is an efficient speaker adaptive technique where features are re-normalized to a particular speaker's domain.  The rest of the machinery is well motivated and well executed, but less novel.  This addition enables the voice conversion between speakers.

Section 4.4 Are all of the utterances the same length?  Based on the architecture description, it appears as though the model generates one output frame for each input frame.  This would suggest that for training input and output need to be synchronized.  If so, make this explicit and include length parameters in Section 6.1

Section 6.2 states "For statistically significant analysis, results are shown in different conversion possibilities." However, no test of statistical significance is presented.  This pointer may be helpful (https://pdfs.semanticscholar.org/b2b1/d01336323f3794f54de26567335aa0bcac46.pdf)

Presentation Comments:

Section 3.1: I would recommend using different subscripts for Z_i and U_i, since when indexing Z this implies the i-th speaker, and when indexing U it's the i-th utterance.  The formulas in Section 3.1 imply a single index i for both of these which is clearly not intended.

Section 4: Consider using the present tense instead of the perfect tense when describing the results.  "...we discuss our proposed AdaGAN architecture... We have shown... We have presented..." can be "...we discuss our proposed AdaGAN architecture... We show... We present..."

Section 5.2; Tables 1 and 2: Consider some partition of the FLOPS and Parameters, separation by commas, spaces or even abbreviation e.g. 2952233 -> 2,952,233 or 2 952 233 or 2.9M.  This will make this table much easier to read.

Section 6.2; Figures 2-5: MOS scores have a minimum value of 1.  This should be the axis of the chart, rather than 0.

It's pretty bold to star by contextualizing the work with the sentence "Language is the core of civilization and speech is the most powerful and natural form of communication."  :-)

**Experience Assessment:**

I have read many papers in this area.

**Review Assessment: Checking Correctness Of Derivations And Theory:**

I assessed the sensibility of the derivations and theory.

**Review Assessment: Checking Correctness Of Experiments:**

I carefully checked the experiments.

**Review Assessment: Thoroughness In Paper Reading:**

I read the paper thoroughly.

---

> ### Author Response · Authors · 2019-11-14
> **Response to Reviewer #1**
>
> We would like to thank respected Reviewer 1 for their reviews and constructive suggestions. We are glad that the reviewer liked our work. Below, we provide clarification for the reviewer’s queries.
>
> Q 1. Section 4.4 Are all of the utterances the same length? Based on the architecture description, it   appears as though the model generates one output frame for each input frame. This would suggest that for training input and output need to be synchronized. If so, make this explicit and include length parameters in Section 6.1
>
> Ans: Thanks for this query. No, the length of utterances is not the same. Based on the length of the utterances, the number of frames will change. Moreover, our model generates one output frame for each input frame. Due to this variable utterances, Dynamic Time Warping (DTW) or any other methodologies are used for synchronization on parallel-data. However, proposed AdaGAN is designed for non-parallel data. Hence, there is no need to do this kind of synchronization. Thank you for your query about length of utterance, we have updated the section 6.1 accordingly to clarify this point.
>
> Q 2. Section 6.2 states "For statistically significant analysis, the results are shown in different conversion possibilities." However, no test of statistical significance is presented. This pointer may be helpful.
>
> Ans: Thanks for this suggestion. We have shown the margin of error corresponding to the 95% confidence interval to quote the statistical significance of our results. In addition, we have incorporated p-value for the subjective test to present the statistical significance of the results. In particular, we obtain 0.013 p-values for the subjective test, which is less than 0.05. This indicates the significance of the results. Thanks again!
>
> Q 3. Section 3.1: I would recommend using different subscripts for $Z_i$ and $U_i$, since when indexing $Z$ this implies the $i^{th}$ speaker, and when indexing $U$ it's the $i^{th}$ utterance. The formulas in Section 3.1 imply a single index i for both of these which is clearly not intended.
>
> Ans: Thanks for this suggestion. However, we used $i$ as the common index. $Z$ is used for speaker, and $U$ for utterance. Hence, $Z_i$ indicates $i^{th}$ speaker, and $U_i$ indicates $i^{th}$ utterance. For example, the index is the same for the first speaker, i.e., $Z_1$ and first utterance, i.e., $U_1$. Thanks again.
>
> Q 4. Section 4: Consider using the present tense instead of the perfect tense when describing the results. "...we discuss our proposed AdaGAN architecture... We have shown... We have presented..." can be "...we discuss our proposed AdaGAN architecture... We show... We present..."
>
> Ans: Thank you for your guidance. We will be careful from now onwards regarding this. Also, we have made the changes (using present tense instead of the perfect tense) in section 1, 4, and 6 accordingly in the updated manuscript.
>
> Q 5. Section 5.2; Tables 1 and 2: Consider some partition of the FLOPS and Parameters, separation by commas, spaces or even abbreviation e.g. 2952233 -> 2,952,233 or 2 952 233 or 2.9M. This will make this table much easier to read.
>
> Ans: Thank you for this valuable suggestion. We have updated the number representations in revised manuscript.
>
> Q 6. Section 6.2; Figures 2-5: MOS scores have a minimum value of 1. This should be the axis of the chart, rather than 0.
>
> Ans: Yes, we strongly agree with the respected reviewer. Thanks. We have made changes in range of the y-axis from 1 to 5 in the revised manuscript.

---

### Public Comment · ~Ju-Chieh_Chou1 · 2019-09-30
**About related work**

Hi,
Thank you for interesting work.
I am the author of this paper: https://arxiv.org/abs/1904.05742
I found that we adopted similar idea (adaIN) to the task of VC.
I believe that including my work in your paper can make your work more thorough.

---

> ### Author Response · Authors · 2019-10-14
> **Thank you for sharing your work!**
>
> Hello,
> Again, thank you for reading our research work and drawing our attention to your nice piece of work.
> We will definitely look into this.

---

### Decision · Program_Chairs · 2019-12-19

**Decision:**

Reject

**Comment:**

The paper has major presentation issues. The rebuttal clarified some technical ones, but it is clear that the authors need to improve the reading substantially, ,so the paper is not acceptable in its current form.